# Efflux-Related Carbapenem Resistance in *Acinetobacter baumannii* Is Associated with Two-Component Regulatory Efflux Systems’ Alteration and Insertion of ΔAbaR25-Type Island Fragment

**DOI:** 10.3390/ijms24119525

**Published:** 2023-05-31

**Authors:** Alicja Słoczyńska, Matthew E. Wand, Lucy J. Bock, Stefan Tyski, Agnieszka E. Laudy

**Affiliations:** 1Department of Pharmaceutical Microbiology and Bioanalysis, Medical University of Warsaw, 02-097 Warsaw, Poland; alicja.sloczynska@gmail.com; 2UK Health Security Agency, Research and Evaluation, Porton Down, Salisbury SP4 0JG, UK; matthew.wand@ukhsa.gov.uk (M.E.W.); lucy.bock@ukhsa.gov.uk (L.J.B.); 3Department of Antibiotics and Microbiology, National Medicines Institute, 00-725 Warsaw, Poland; s.tyski@nil.gov.pl

**Keywords:** multidrug resistance, carbapenem resistance, efflux pump, insertion sequence, resistance island, EPI, two-component regulatory system, AdeABC

## Abstract

The efflux pumps, beside the class D carbapenem-hydrolysing enzymes (CHLDs), are being increasingly investigated as a mechanism of carbapenem resistance in *Acinetobacter baumannii*. This study investigates the contribution of efflux mechanism to carbapenem resistance in 61 acquired *bla*_CHDL_-genes-carrying *A. baumannii* clinical strains isolated in Warsaw, Poland. Studies were conducted using phenotypic (susceptibility testing to carbapenems ± efflux pump inhibitors (EPIs)) and molecular (determining expression levels of efflux operon with regulatory-gene and whole genome sequencing (WGS)) methods. EPIs reduced carbapenem resistance of 14/61 isolates. Upregulation (5–67-fold) of *adeB* was observed together with mutations in the sequences of AdeRS local and of BaeS global regulators in all 15 selected isolates. Long-read WGS of isolate no. AB96 revealed the presence of AbaR25 resistance island and its two disrupted elements: the first contained a duplicate IS*Aba1-bla*_OXA-23_, and the second was located between *adeR* and *adeA* in the efflux operon. This insert was flanked by two copies of IS*Aba1*, and one of them provides a strong promoter for *adeABC*, elevating the *adeB* expression levels. Our study for the first time reports the involvement of the insertion of the ΔAbaR25-type resistance island fragment with IS*Aba1* element upstream the efflux operon in the carbapenem resistance of *A. baumannii*.

## 1. Introduction

*Acinetobacter baumannii* is a non-fermentative, Gram-negative coccobacillus, responsible for numerous opportunistic and nosocomial infections. It contributes to a variety of disease states, including ventilator-associated pneumonia (VAP), meningitidis, bloodstream, skin and soft tissues infections. It has been implicated in wound and surgical site contamination, as well as contributing to catheter-borne infections amongst others [1]. During of the SARS-CoV-2 (COVID-19) pandemic, carbapenem-resistant *A. baumannii* (CRAB) was a leading cause of VAP in SARS-CoV-2 infected patients [2]. *A. baumannii* is resistant to treatment by many antibiotics due to its intrinsic and acquired resistance mechanisms. Current treatment strategies include certain aminoglycosides, tigecycline, colistin, β-lactams with sulbactam and the “last-resort” antibiotics—carbapenems [3]. Of major concern is the increased incidence of carbapenem resistance, which has risen dramatically over the last few decades. According to the European Centre for Disease Prevention and Control report, in 2020, laboratories from 29 EU/EEA (European Union/European Economy Area) countries reported the detection of 7622 invasive isolates belonging to the genus *Acinetobacter*, of which 7542 (99%), showed resistance to carbapenems [4]. The multidrug resistance (MDR) of the phenotype of hospital strains to a wide spectrum of antibiotics and chemotherapeutic agents, including carbapenems, is now a very significant public health problem. In the last six years, only two new drugs (eravacycline and cefiderocol—siderophore cephalosporin) which showed activity against *A. baumannii* have been approved by the European Medicines Agency (EMA) [5].

Intrinsic carbapenem resistance mechanisms in *A. baumannii* include production of chromosomally encoded carbapenemases [6], decreased outer membrane permeability [7] and the presence of various efflux systems [7] combined with genomic plasticity and ability to easily adapt to more demanding conditions by modulating gene expression can give rise to an extensively drug-resistant isolates (XDR) with no therapeutic options [8]. By far, the most common carbapenemases in *A. baumannii* are the carbapenem-hydrolysing class D enzymes (CHDLs). The CHDL enzyme families identified in *A. baumannii* are intrinsic OXA-51-like and acquired OXA-23-like, OXA-24-like, OXA-58-like, OXA-143-like and OXA-235-like [6,9]. Apart from OXA-type enzymes, *A. baumannii* strains have been found to contain plasmids carrying other carbapenemases, such as metallo-β-lactamases (MBLs), e.g., NDM-1 or class A serine carbapenemases including *Klebsiella pneumoniae* carbapenemases (KPCs) [6]. However, similar to other Gram-negative rods, carbapenem resistance is often a result of multiple resistance mechanisms including efflux pumps [10].

The following superfamilies of multidrug resistant (MDR) efflux pumps have been identified in *A. baumannii* so far: RND (resistance-nodulation-cell division), MATE (multidrug and toxic compound extrusion), MFS (major facilitator superfamily) and SMR (small multidrug resistance) [11]. Pumps belonging to RND family are ubiquitous in *A. baumannii* and demonstrate the broadest substrate range, including antibiotics, chemotherapeutics, non-antibiotic drugs, dyes, biocides, detergents and antiseptics [11,12]. The overproduction of the MDR efflux pumps is associated with drug resistance. AdeABC, AdeIJK and AdeFGH are the three major RND efflux systems which have been identified in *A. baumannii* [13]. It has been shown that overproduction of AdeABC is responsible for reduction of susceptibility to aminoglycosides, fluoroquinolones and tigecycline amongst clinical strains [14,15]. Its substrate range also encompasses chloramphenicol and β-lactams, including carbapenems [10,13]. This efflux system is composed of a tripartite proteins assembly spanning the inner and outer membranes: the AdeA membrane fusion component, the AdeB multidrug transporter and the AdeC outer membrane component [16]. Amongst the three mentioned RND efflux systems, only AdeABC has been implicated in carbapenem resistance [10,17,18]. AdeABC as well as other efflux systems are encoded by genes organized in operons that are located in bacterial chromosomes [13].

Two-component regulatory systems (TCSs) function as an important mechanism which enables bacteria to recognize, response and adapt to various environmental stimuli [10,14,15,16,19]. AdeRS is a local TCS which regulates the expression of the *adeABC* operon [10,14,16]. The gene encoding AdeRSs are located immediately upstream and are divergently transcribed to *adeABC* [16]. AdeS is the sensor kinase which, in response to environmental stimulus autophosphorylates, transfers the phosphate group to the response regulator AdeR [16]. Between *adeRS* and *adeABC* operons, there is a 133 bp intercistronic spacer with a direct-repeat motif to which AdeR binds, regulating expression of *adeABC* [20]. Changes in *adeRS*, including substitutions, deletions and insertions may lead to overexpression of *adeABC* operon [21]. However, AdeABC overproduction is not determined only by the mutations within AdeRS sequences. Lin et al. [19] suggested that regulation of *adeABC* expression is controlled by another TCS—BaeSR. It is a global TCS and responses to environmental stress stimuli. It was shown that BaeSR contributed to regulation of *adeABC* expression by influencing susceptibility to tigecycline, an antibiotic whose susceptibility is also governed by *adeABC* overexpression [22]. However, the relationship between BaeSR and AdeABC, is not yet clear and needs further investigation.

AbeM (MATE family) was suggested as another efflux pump which may influence carbapenem susceptibility [23]. Apart from carbapenems, it also has fluoroquinolones, gentamicin, doxorubicin and triclosan within its substrate range [24]. AbeM is a prevalent pump in *A. baumannii* and can be found in up to 98% of isolates [25].

In our previous paper, we performed phenotypic and molecular characterisation of the collection of 61 imipenem-non-susceptible *A. baumannii* clinical isolates, focusing on carbapenemases and insertion sequences (ISs). That study revealed the presence of *bla*_OXA-24-like_ in 39/61 isolates_,_ IS*Aba1*-*bla*_OXA-23-like_ in 14/61 isolates and IS*Aba3*-*bla*_OXA-58-like_ in 6/61 isolates [9]. The location of an IS upstream of the *bla*_CHDL_ gene should lead to the overproduction of CHDL enzyme [26]. However, when performed on strains where the IS preceded *bla*_CHDL_, the CarbAcineto NP Test revealed non-obvious and uninterpretable levels of carbapenemase activity. High carbapenem MIC values together with the uninterpretable CarbAcineto NP test results for the isolates with acquired *bla*_CHDL_ genes suggested additional resistance mechanisms affecting carbapenem activity and prompted us to investigate the contribution of efflux pumps [9].

The objective of this study was to determine the impact of the most important efflux system, AdeABC, and additionally the AbeM pump on carbapenem resistance in clinical isolates of acquired *bla*_CHDL_-gene-carrying *A. baumannii*. The goal was to be achieved by relying on phenotypic (susceptibility testing to carbapenems with efflux pump inhibitors (EPIs)) and molecular methods (determining expression levels of *adeB* pump gene with *adeSR* regulatory genes, whole genome sequencing and analysing changes of the following genes: *abeM*, *baeSR* and *adeSRABC,* with special attention to genetic environment of the latter).

## 2. Results

### 2.1. Susceptibility Profiles and EPI Effects on Carbapenem Resistance

The susceptibility results of the tested 61 isolates to eight antibiotics (meropenem, imipenem, cefepime, ceftazidime, ciprofloxacin, levofloxacin, gentamicin and tobramicin) that are substrates of the AdeABC and AdeM efflux pumps are presented in Appendix A. The results showed that 100% of isolates were resistant to meropenem, and 98.4% (60/61) to imipenem. One isolate was intermediately resistant to imipenem. At the same time, 93.4% isolates showed resistance to ceftazidime, and the remaining 6.6% showed intermediate susceptibility to it. More than 50% were susceptible and/or intermediately susceptible to cefepime. All the isolates were ciprofloxacin resistant, and about 50% showed resistance to levofloxacin. About 75% of the isolates were resistant to aminoglycosides gentamicin and tobramycin.

The MIC values of imipenem and meropenem were evaluated for all 61 isolates in the presence and the absence of two efflux pump inhibitors, Phe-Arg-β-naphthylamide (PAβN) and Carbonyl cyanide 3-chlorophenylhydrazone (CCCP), to determine the potential role of efflux pumps in carbapenem resistance (Table 1, Appendix A). At least a 4-fold reduction in meropenem MIC values in the presence of PAβN was observed for 21% (13/61) of studied isolates, whereas the addition of CCCP did not produce any effect on meropenem MIC in any of the isolates tested. Over 50% (7/12) of isolates which exhibited a meropenem MIC reduction in the presence of PAβN carried the IS*Aba1*-*bla*_OXA-23-like_ gene. The remaining PAβN-affected strains consisted of one IS*Aba3*-*bla*_OXA-58-like_-carrying isolate and four *bla*_OXA-24-like_-carrying isolates. Imipenem MIC values were significantly reduced in the presence of PAβN in only three isolates (two isolates with a IS*Aba1*-*bla*_OXA-23-like_ gene and one with a *bla*_OXA-24-like_ gene). CCCP reduced imipenem MICs in only two isolates, one isolate belonged to the *bla*_OXA-24-like_ group and one isolate to the IS*Aba3*-*bla*_OXA-58-like_-possessing group. It should be emphasized that only 6.6% (4/61) of isolates demonstrated simultaneous imipenem and meropenem MIC reduction in the presence of at least one inhibitor. A summary of the MIC reduction of both carbapenems after the addition of EPIs and the presence of genes encoding the acquired CHDL enzymes and the activity of carbapenemases in the CarbAcineto NP assay [9] for all 61 tested isolates are presented in Appendix A.

### 2.2. Presence and Expression of Efflux Pump Genes

PCR was used to show that all 61 *A. baumannii* clinical isolates contained genes coding the AdeB and AbeM efflux pumps.

In order to determine the contribution of carbapenemase and efflux activity upon carbapenem resistance, three aspects of the *A. baumannii* isolates were taken into account when selecting isolates for further analysis: OXA-type production, CarbAcineto NP test result [9] and EPIs impact on the meropenem MIC value. This resulted in a group of 15 isolates selected for further testing. Isolates were divided into three groups depending on the type of *bla*_OXA_ gene carriage: (I) IS*Aba3*-*bla*_OXA-58-like_ (the number of selected isolates is two (*n* = 2), (II) IS*Aba1*-*bla*_OXA-23-like_ (*n* = 7), (III) *bla*_OXA-24-like_ (*n* = 6). Each group consisted of isolates with and isolates without at least a 4-fold reduction in meropenem MIC value observed in the presence of PAβN.

To define the role of AdeABC and its regulators AdeRS in carbapenem resistance in the selected 15 isolates, expression of *adeB, adeR* and *adeS* was analysed by quantitative PCR (qPCR). For each isolate, a significant (at least a 3-fold) upregulation of *adeB* was observed (from 5- to 67-fold) compared with *A. baumannii* ATCC 17978 which was used as a baseline (Table 2). Isolates with the highest expression level of *adeB* (49-, 58- and 67-fold) all belonged to the IS*Aba1-bla*_OXA-23_-_like_ group (isolate nos. AB129, AB185 and AB96, respectively), whilst no upregulation (>2-fold) of *adeSR* regulatory genes was observed. However, for two isolates (no. AB87 and AB165), a significant (between three- and four-fold) upregulation of *adeR* was observed (Table 2). Moreover, in the case of seven other isolates (nos. AB43, AB86, AB118, AB76, AB81, AB159 and AB176), a 2-fold increase in *adeR* expression was obtained. Only one isolate (*A. baumannii* 165) showed upregulation of the *adeS* gene expression.

### 2.3. Amino Acid Sequence analysis of Efflux Pumps and their TCS Regulators

All 15 isolates were short-read whole genome sequenced (WGS) and screened for mutations of *adeRS*, *adeA*, *adeB*, *baeS*, *baeR* and *abeM*. Carbapenem-sensitive *A. baumannii* ATCC 17978 was used as a reference for sequence comparison and identification of single-nucleotide-polymorphisms (SNPs). Sequence analysis of *adeSR* revealed nucleotide changes in *adeS* in all isolates compared to the reference isolate, which resulted in changes in 7 amino acids located in positions 172, 186, 268, 303, 7348, 356 and 357 (Table 2, Figure 1 and Appendix A). Furthermore, all isolates were missing four terminal amino acids: E358, E359, I360 and G361. In addition, in three isolates (nos. AB96, AB129 and AB185), there was a deletion of two amino acids (H102-G103) in AdeS.

The same three isolates nos. AB96, AB129 and AB185 have an IS*Aba1* insertion within *adeR* gene sequence, which results in a truncated AdeR protein. All 15 isolates had SNPs mutations leading to the amino acid changes in positions 120 and 136 in AdeR sequence when compared to *A. baumannii* ATCC 17978. Isolate no. AB165 had an additional amino acid substitution in position 115 (Table 2). The nucleotide sequence of *adeR* of isolate no. AB165 in comparison to the reference strain *A. baumannii* ATCC 17978 is presented in Figure 2.

Nucleotide sequence analysis of *adeAB* revealed changes in the amino acid sequence of AdeA in three positions K25E, K352R and T391A in each studied isolate. As for the global TCS, a mutation was found that caused one change in amino acid position S437T in BaeS in all isolates. There were no changes in amino acid sequence of AdeB and BaeR in any of the clinical isolates. In addition, only *A. baumannii* 185 had truncated AbeM protein due to mutation L434STOP. The whole genome datasets of the 15 strains generated and analysed during the current study were previously deposited in the NCBI GeneBank (Submission ID: SUB9082120, BioProject ID:PRJNA701882).

### 2.4. Core-Genome SNP Phylogenetic Analysis

To explore more how closely are related all the selected 15 *A. baumannii* isolates, they were genotyped by the core-genome SNP-based phylogenetic analysis. This study revealed the selected isolates to form three clusters (Figure 3). Due to the obtained results of the short-read WGS analysis and thus the similarities of changes in the *adeSR-adeABC* region, the most interesting were isolates nos. AB96, AB129 and AB185. The use of *A. baumannii* 96 as a reference allowed the most visible graphical presentation of the genetic diversity of the tested isolates and showed that it clusters together with isolates nos. AB129 and AB185, all of which showed the same nucleotide sequence changes in *adeRS*.

### 2.5. Analysis of Genetic Structure of adeRS and adeABC Region in Isolate A. baumannii 96

Short-read WGS and bioinformatic analysis of *adeSR-adeABC* operon in all 15 isolates revealed disruption of this region, resulting in truncated *adeR* and an IS*Aba1* insertion upstream of *adeA* in the three isolates from the same cluster (nos. AB96, AB129 and AB185). *A. baumannii* 96 was selected for further genetic analysis using nanopore sequencing. This isolate showed the highest upregulation of the *adeB* gene (67-fold) and resistance to both carbapenems (MIC = 32 mg/L for meropenem and imipenem) and a reduction (4-fold) in the MIC value of imipenem in the presence of PAβN. The comparison of the obtained final genomic sequence of isolate no. AB96 with the database of sequences deposited in NCBI GenBank revealed the presence of a complete sequence of a resistance island AbaR25 (46.4 kb) and located elsewhere in the genome, two disrupted elements of mentioned island: 13.5 kb long first element which will now be referred to as Δ1AbaR25-fragment and 16.5 kb long second element referred to as Δ2AbaR25-fragment. The first element of this island, Δ1AbaR25-fragment, was found in the region between the *adeR* and *adeA* genes (Figure 4). This insertion resulted in both a truncated AdeR protein and loss of the greater part of intercistronic spacer with the AdeR binding site. The IS*Aba1* located upstream of *adeA* was oriented in the opposite direction providing a strong promoter for the *adeABC* operon. Comparison of Δ1AbaR25-fragment with AbaR25 revealed several nucleotide changes” a substitution within *tniC* which resulted in new protein TniC_1_ (E174K), double nucleotide deletion ΔTC within *tniA* which resulted in the presence of two open reading frames *tniA*Δ_1_ and *tniA*Δ_2_, single nucleotide deletion ΔC within *tniE* which resulted in a truncated TniE protein (ΔTniE_)_, double nucleotide deletion ΔAG within *sup* which resulted in two open reading frames *sup*Δ_1_ and *sup*Δ_2_. The second element of this island, Δ2AbaR25-fragment, was present in another place in the genome of isolates no. AB96. This fragment had Tn*2006* carrying *bla*_OXA-23-like_ gene, which led to the duplication of the *ISAba1-bla_OXA_*_-23-like_ region in isolate no. AB96’s genome. This genome sequence is available at the NCBI BioProject repository (Submission ID: SUB12923155, BioProject ID: PRJNA939738).

## 3. Discussion

Previous studies have shown that carbapenems are substrates for two pumps AdeABC and AbeM in *A. baumannii* [18,23]. Roy et al. confirmed the ability of several carbapenems (imipenem, meropenem, doripenem and ertapenem) to bind to the active site in AdeB using a sequence-structure based in-silico approach [10]. The ability of EPI PAβN to bind to AdeB and its activity as a competitive inhibitor of this efflux pump were also confirmed [10]. Our study sheds new light on the involvement of efflux pumps in carbapenem resistance in *A. baumannii*, regardless of the presence and activity of various acquired CHDL enzymes. All 61 isolates used in this study were previously phenotypically and molecularly characterized for carriage of various acquired *bla*_CHDL_ genes [9]. The contribution of efflux to carbapenem resistance was investigated in a subset of these strains in this study. Phenotypic tests detecting the contribution of efflux pumps in drug resistance in several Gram-negative bacteria use EPIs such as PAβN and CCCP [27,28,29,30,31]. At least a 4-fold reduction in antibiotic MIC values in the presence of EPI indicates efflux-related resistance [27,30,31]. PAβN is a specific competitive inhibitor for efflux pumps belonging to the RND family [10,23]. The second inhibitor CCCP is widely and successfully used in *Acinetobacter* efflux studies with different pumps substrates such as tigecycline, ciprofloxacin, but also meropenem [32,33,34] and imipenem [35,36]. In our study the reduction of carbapenem MIC in the presence of EPIs was observed for only a few isolates, regardless of the acquired CHDL enzymes they produced. The production of CHDL carbapenemases was previously confirmed by the CarbAcineto NP test for all of these isolates, except for the OXA-58-positive isolate no. AB43 [9]. It may suggest that increased efflux is the leading carbapenem resistance mechanism in this isolate.

AdeABC, belonging to RND family, is the most important MDR efflux pump in *A. baumannii*. Its substrate range is extensive and encompasses many substances including several β-lactams [37,38,39]. Its role in carbapenem resistance is still unclear since resistance to carbapenems is mainly associated with carbapenemase production [6]. Previous studies indicate that efflux activity may contribute to carbapenem resistance in *A. baumannii* [14,23,29]. Zhang et al. reported that a imipenem-selected stress in *A. baumannii* strain led to overproduction of AdeABC and consequently to the reduction of susceptibility to a variety of antibiotics including carbapenems [18]. Moreover, the *adeB* expression levels were upregulated in *A. baumannii* isolates with a reduction in the MIC of meropenem in the presence of PAβN [29]. However, Salehi et al. did not observe imipenem MIC reduction in the presence of PAβN in *A. baumannii*, even though expression of *adeB* was significantly upregulated [40]. In our study, all 15 isolates, chosen for *adeB* expression level analysis, had significant (≥3-fold) upregulation of this gene, regardless of PAβN effect on meropenem MIC and the CarbAcineto NP result. However, three isolates nos. AB96, AB129 and AB185 (each possessing an IS*Aba1*-*bla*_OXA-23-like_) presented noticeably the highest *adeB* expression, up to 67-fold upregulation. A significant reduction of the meropenem MIC value in the presence of PAβN was observed only for *A. baumannii* 96, while for the other two isolates, there was no difference. The isolate no. AB96 was the only one of the three where a positive CarbAcineto NP test result was observed, suggesting higher carbapenemase activity for this isolate. On the other hand, it was presented by Salehi et al. that despite the lack of influence of PAβN on carbapenem MIC values, upregulation of *adeABC* in *A. baumannii* was observed [40].

The most common cause of *adeABC* overexpression are changes in nucleotide sequences of regulatory genes. To date, both the local and global regulators of the AdeABC efflux system have been identified. BaeSR is a global regulatory system which was showed to regulate expression of *adeA* and *adeB* genes [19]. One mutation found within BaeS amino acid sequence, S437T, was associated with upregulation of *adeB* gene by Salehi et al. [40]. It is worth noting that all of 15 isolates analysed in our study were positive for this mutation, and simultaneously all of them presented overexpression of *adeB,* while Salehi et al. reported only one isolate with this mutation. However, as with the efflux pumps of other Gram-negative bacilli, the influence of local regulators is better documented. Various studies reported that specific mutations in the *adeRS* TCS local regulatory system of AdeABC resulted in overproduction of AdeB leading to decreased antimicrobial susceptibility [18,21]. Within AdeS, the following substitutions of amino acid sequence were observed: T153M in histidine box, D30G in periplasmic loop, G186V in the αhelix of the dimerization and DHp domain, G103D in HAMP linker domain [14,37,41,42]. An IS*Aba1* insertion within adeS resulting in a truncated AdeS may also be responsible for enhancing expression of efflux pump operons [21]. In all 15 sequenced isolates, we have identified in AdeS a G186V point mutation that can alter the conformation of AdeS DHp domain and then restore expression [43,44]. However, the other observed six amino acid substitutions (L172P, N268H, Y303F, V348I, G356N, H357N) and four terminal amino acid deletions in AdeS considered as may-be silent, not associated with the overexpression of *adeABC* operon. Furthermore, three isolates (*A. baumannii* numbers 96, 129 and 185) were missing H102 and G103 in the HAMP linker domain of AdeS. This, to our knowledge, is the first deletion observed at this location in AdeS, although the substitution G103D has been reported before [45]. Compared to AdeS, changes in the amino acid sequence of AdeR were less frequently observed. Mutations in AdeR which can boost *adeB* expression are D20N, D26N, A136V in the phosphorylation site, A91V of signal-receiving domain and P116L at the first residue of helix α5 [18,37,41,46]. Sequence comparison of our tested strains revealed two mutations, V120I and A136V, in AdeR in all of them. Haeili et al. reported both of these substitutions in isolates with elevated *adeABC* pump expression [44].

Most importantly, our short-read WGS results revealed that the highest levels of upregulation of *adeB* in the three isolates (nos. AB96, AB129 and AB185) were due to the presence of additional IS*Aba1* sequences both upstream of *adeA* and into *adeR*. An IS*Aba1* insertion identified upstream of *adeA* provides a strong promoter for *adeABC* operon. On the other hand, an IS*Aba1* insertion starting downstream of *adeR* resulted in disruption of the gene and forming a truncated protein. To date, only one *A. baumannii* clinical strain (no. Ab209) has been reported to have an insertion of IS*Aba1* upstream of the *abaABC* operon [47]. The authors concluded that IS*Aba1* governs the expression of the *adeABC* operon despite disruption of *adeR* by the second IS*Aba1* in the same isolate. This is because IS*Aba1* provides a strong promoter upstream of the gene *adeA*. However, unlike the Polish isolates with upregulation from 49- to 67-fold, only 2.56-fold upregulation of the *adeB* gene of strain Ab209 compared to *A. baumannii* ATCC 19606 was obtained [47]. Moreover, Zang et al. showed that IS*Aba1* insertion upstream membrane fusion protein AdeI encoding gene increased transcription of a different RND efflux pump operon *adeIJK* [48]. However, in our three isolates, the rest of the *adeR-adeA* region remained unknown; therefore, a long-read WGS was performed. Finally, the obtained complete sequence of isolate no. AB96’s genome revealed the presence of the AbaR25 resistance island described previously [49] and located elsewhere in the genome and two disrupted fragments of this island. It should be emphasized that AbaR-type islands (AbaRs) are important genetic elements responsible for antimicrobial resistance in *A. baumannii* strains [50,51]. The AbaR25 belongs to the AbaR4-like group of resistance islands that carry the most important IS*Aba1*-*bla*_OXA-23_ gene and also other resistance genes such as *sul2*, *tetA*, *strA* and *strB*. The island of AbaR25 was located in chromosome of isolate no. AB96 as most of the previously documented cases [50].

Most importantly, the unknown region *adeR-adeA* of the tested isolate no. AB96 turned out to be a 13.5 kb-long element of the AbaR25 island (the Δ1AbaR25-fragment), which was flanked by two copies of IS*Aba1*. We suppose that this fragment of the AbaR25 island was most likely duplicated in the genome of isolate no. AB96 by homologous recombination. This insertion resulted not only in truncated AdeR protein but also in the loss of the intercistronic spacer; an AdeR binding site located between *adeR* and *adeA*. Chang et al. demonstrated that a direct repeat motif (“AAGTGTGGAG” separated with an “A” nucleotide) is the minimal element to enable binding of AdeR [20]. As this region was deleted in *A. baumannii* 96, AdeR is unable to regulate *adeABC* operon expression. Taking the above into account, the location and orientation of the second IS*Aba1* sequence flanking the island fragment became crucial for *abaABC* expression in tested isolate no. AB96. It is known that IS*Aba1* provides a strong promoter with both -10 and -35 sites located within IS sequence. However, its orientation is crucial to actually elevate expression [52]. The nucleotide sequence analysis of *A. baumannii* 96 genome revealed the presence of a strong promoter with an extended –10 motif “TGACATTATTT”, indicating that it can play a significant role in controlling the expression of *adeABC*. It is the first time the insertion of a fragment of the resistance island AbaR25 in the region between *adeR* and *adeA* is detected and analysed regarding the *adeB* overexpression.

Interestingly, apart from the AbaR25 resistance island itself and the Δ1AbaR25-fragment, an additional second element of AbaR25 (Δ2AbaR25-fragment) was found in the genome of isolate no. AB96. It contained a Tn2006 carrying IS*Aba1-bla*_OXA-23-like_ gene. IS*Aba1* provides a strong promoter, elevating the *bla*_OXA-23-like_ expression levels [53].

It is generally believed that the resistance of acquired *bla*_CHDL_-genes-carrying strains results from the production of CHDL enzymes, especially when dealing with the presence of a strong promoter as is the case in IS*Aba1-bla*_OXA-23-like_ gene. Moreover, the wide prevalence in *A. baumannii* clinical strains of AbaR4-like resistance islands contributes to the presence of the IS*Aba1-bla*_OXA-23-like_ gene. Our study for the first time reports that despite the duplication of the IS*Aba1-bla*_OXA-23_ gene in the genome, the insertion of an AbaR25-island fragment with the IS*Aba1* sequence as a strong promoter of the *adeABC* operon is important for the contribution of the efflux system to carbapenem resistance.

## 4. Materials and Methods

### 4.1. Bacterial Strains

A collection of 61 non-repetitive imipenem-insensitive *A. baumannii* strains were isolated in the period 2009–2014 from patients hospitalised in one tertiary hospital in Warsaw, Poland. The isolates were recovered from the following clinical specimens: respiratory tract samples (*n* = 15), wound swabs (*n* = 15), urine samples (*n* = 15), rectal swabs (*n* = 4), blood samples (*n* = 3), fistula swabs (*n* = 3), stoma swabs (*n* = 3) and one isolate from each, peritoneal fluid, surgical drain swab and catheter tip. All studied strains were characterized (clinical material type, antimicrobial susceptibility determination, phenotypic carbepenemase detection, presence of genes encoding carbapenemases, genetic relatedness between strains by PFGE method, MLST sequence type) in our previous paper [9]. All strains carried *bla*_CHDL_ genes relevant for carbapenem resistance, including 59 isolates with acquired *bla*_CHDL_ genes and the other 2 isolates with IS*Aba1-bla*_OXA-51_-_like_ genes [9]. Epidemiological data of isolates and the *bla*_CHDL_-gene groups they possess are presented in Appendix A. All strains were stored at −80 °C in LB broth supplemented with glycerol until analysis. Prior to testing, each strain was sub-cultured on tryptic soy agar (TSA) (bioMérieux, Mercy l’Etoile, France) medium for 24 h at 35 °C to ensure viability.

### 4.2. Determination of the MICs of AdeABC and AbeM Efflux Pumps Substrates

Minimal inhibitory concentrations (MICs) of carbapenems (imipenem and meropenem), cephalosporins (cefepime and ceftazidime), fluoroquinolones (ciprofloxacin and levofloxacin), and aminoglycosides (gentamicin and tobramicin), were determined using the VITEK 2 system (bioMérieux, Mercy l’Etoile, France). Interpretation of the MIC results was performed according to CLSI breakpoints [54].

### 4.3. Determination of the MICs of Carbapenems with and without Efflux Pump Inhibitors

In order to determine strains with active efflux pumps extruding carbapenems, the MIC values of imipenem and meropenem (both from Sigma, St. Louis, MO, USA) in the presence or absence of efflux pump inhibitors (EPIs) were estimated in Mueller Hinton II (MH II) broth medium (Becton, Dickinson and Company, Franklin Lakes, NJ, USA), using double agent dilutions method, according to the CLSI guideline [55]. The following two RND efflux pump inhibitors (EPIs) Phe-Arg-β-naphthylamide—PAβN and Carbonyl cyanide 3-chlorophenylhydrazone—CCCP (both from Sigma, St. Louis, MO, USA) were used. The final concentrations of PAβN in broth MH II medium were 50 mg/L. Our previous research showed that the MIC values of PAβN were ≥250 mg/L for all tested strains. The second inhibitor CCCP was used at the concentration of one-fourth of MIC of each isolate (i.e., in the range from 0.75 to 5 mg/L). Solutions of PAβN were prepared in deionized water whilst CCCP in dimethyl sulfoxide (DMSO) (Sigma, St. Louis, MO, USA). In order to provide bacterial cell’s outer membrane stabilization, the broth MH II medium was supplemented with 1 mM MgSO_4_ (Sigma, St. Louis, MO, USA) [56]. The plates were incubated for 18 h at 35 °C. At least a 4-fold change in MIC values after addition of EPI was considered significant [30]. Such significant reduction in the MIC values of meropenem or imipenem in the presence of at least one of the EPIs was interpreted as the likely contribution of efflux pumps to carbapenem resistance of the studied isolate.

The assay was validated by the MIC determination of selected antimicrobial agents against reference strain *E. coli* ATCC 25922 and comparison of the experimental MIC values with the CLSI guidelines [54]. The MIC breakpoints for imipenem and meropenem were ≤2, 4 and ≥8 mg/L to designate susceptible, intermediate and resistant strains, respectively [54].

### 4.4. Detection of Genes Encoding AdeABC and AbeM Efflux Pumps 

Detection of *adeB* and *abeM* genes was carried out with a singleplex PCR. The total DNA of the clinical isolates was extracted using a Genomic Mini Kit (A&A Biotechnology, Gdynia, Poland). PCR was performed by using the Hypernova polymerase (Blirt, DNA, Gdańsk, Poland) with the following amplification parameters: 95 °C for 4 min, following 25 cycles of 30 s at 95 °C, annealing 30 s at 55 °C (*adeB* gene) and at 54 °C (*abeM* gene), 45 s at 72 °C and a final extension of 3 min at 72 °C. Sequences of the primers are presented in Table 3.

### 4.5. Whole Genome Sequencing

Genomic DNA of bacterial strains was isolated using a Genomic Mini AX Bacteria Spin Kit (A&A Biotechnology, Gdynia, Poland). Whole genome sequencing was carried out with the following two methods: (I) using Public Health England—Genomic Services and Development Unit (PHEGSDU) on the HiSeq 2500 System (Illumina, Cambridge, UK) with paired end read lengths of 150 bp, for 15 selected isolates, (II) using GridION sequencer (Oxford Nanopore Technologies, Oxford, UK) in the Laboratory of DNA Sequencing and Oligonucleotide Synthesis IBB PAS in Warsaw for one isolate, *A. baumannii* 96. The procedure of short-read WGS by the HiSeq 2500 System was performed as described in our previous paper [9]. To carry out the long-read WGS, the Illumina data of the strain no. AB96 was downloaded from the BioProject no. PRJNA701882. Sequence quality metrics were assessed using FASTQC tool [58] and quality trimmed using fastp [59]. Prior to long-read library preparation genomic DNA was sheared into ~30kb fragments using 26G needle followed by size selection using Short Read Eliminator kit (Circulomics, Baltimore, MD, USA). A total of 3 µg of recovered DNA was taken for 1D library construction using SQK-LSK109 kit, and 0.8 µg of final library was loaded into R9.4.1 flow cell and sequenced on GridION sequencer (Oxford Nanopore Technologies, Oxford, UK). The methodology of the genome assembly of the *A. baumannii* 96 strain is presented in the Appendix A.

Obtained sequences of studied isolates were compared with *A. baumannii* ATCC 17978 isolate with the use of SnapGene, version 6.0.2. Finally using the WGS data, the amino acid changes within *adeABC*, *adeRS*, *baeRS*, *abeM*, were determined. The obtained genomic sequence of isolate no AB96 was compared with those deposited in GenBank by BLASTn.

### 4.6. Core-Genome SNP Phylogenetic Analysis

The Illumina data of the *A. baumanii* strains downloaded from the BioProject no. PRJNA701882 were assembled into contigs using Unicycler. Phylogenetic analysis and SNP tree construction was conducted using kSNP3.0 [60]. Phylogenetic tree was visualized using Figtree [61]. Phylogenetic analysis was performed in cooperation with the Laboratory of DNA Sequencing and Oligonucleotide Synthesis IBB PAS in Warsaw, Poland.

## 5. Conclusions

Our study for the first time reports the involvement of the insertion of the ΔAbaR25-type resistance island fragment with IS*Aba1* element upstream of the efflux operon in the carbapenem resistance of *A. baumannii* clinical isolates. This insert was flanked by two copies of IS*Aba1*, and one of them provides a strong promoter for *adeABC*, elevating the *adeB* expression levels. Interestingly, our study also revealed that AbaR25 resistance island is the reservoir of resistance genes, and they may be transferred to another part of the bacterial genome. Apart from the island of resistance, two of its fragments were found in the chromosome of *A. baumannii* strain: the first contained a duplicate IS*Aba1-bla*_OXA-23_ gene and the second contained an IS element located upstream of the efflux operon. It is worth noting that the analysis of the AdeABC-regulatory genes of the Polish isolate with *adeB*-overexpression showed many point mutations leading to amino acid changes in the AdeRS local regulator as well as in the BaeS global regulator. However, the presence of IS*Aba1* as a strong promoter plays a major role in the high upregulation of *adeABC*. Therefore, our data indicate that in some isolates both efflux systems and CHDL enzymes may participate independently and equally in meropenem resistance.

## Figures and Tables

**Figure 1 ijms-24-09525-f001:**
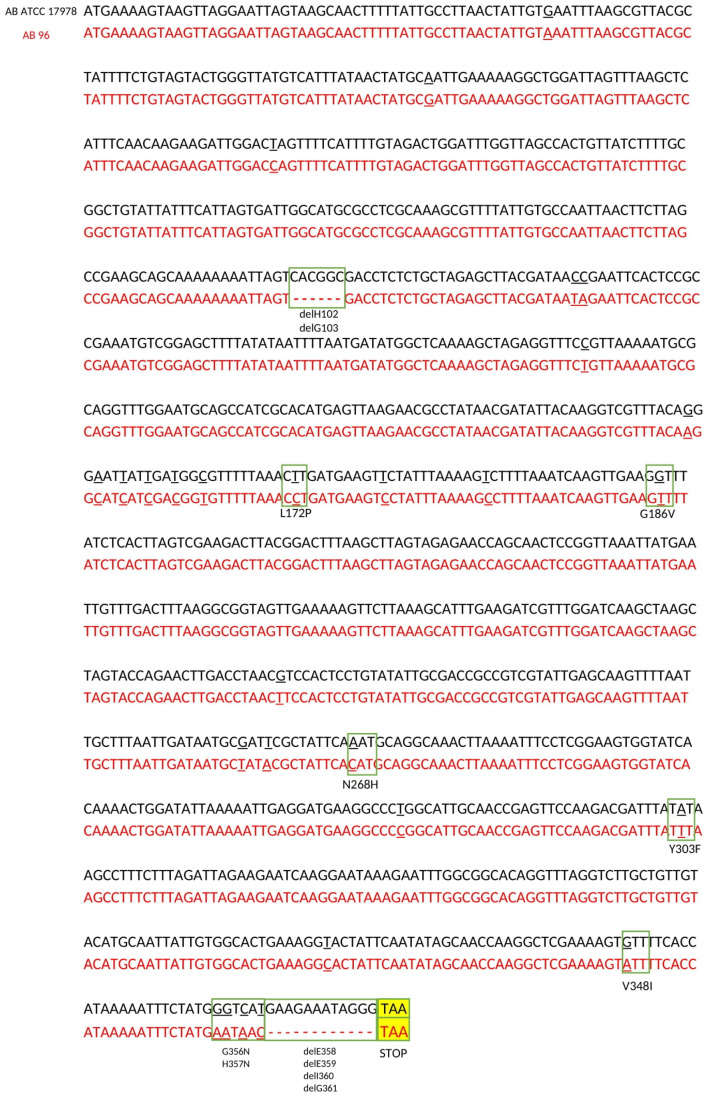
Sequence comparison of the *adeS* gene of clinical isolate *A. baumannii* 96 (in red font) and reference strain *A. baumannii* ATCC 17978 (in black font). The nucleotide positions where the change occurred are underlined. Nucleotide mutations that led to changes in amino acids (substitution and deletion) are shown in the green boxes. The positions of the amino acid changes are given according to the AdeS protein of the reference strain. The STOP codon is shown in the yellow box.

**Figure 2 ijms-24-09525-f002:**
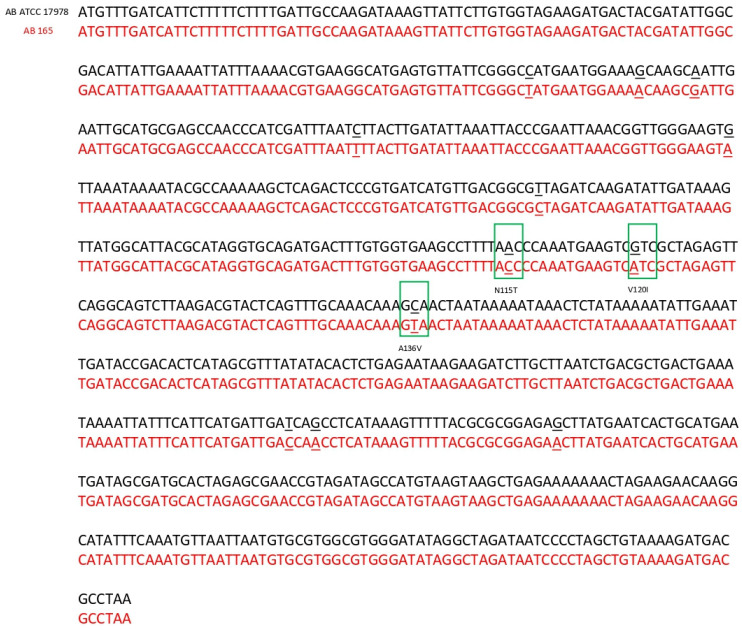
Sequence comparison of the *adeR* gene of clinical isolate *A. baumannii* 165 (in red font) and reference strain *A. baumannii* ATCC 17978 (in black font). The nucleotide positions where the change occurred are underlined. Nucleotide mutations that led to changes in amino acids are shown in the green boxes. The positions of the amino acid changes are given according to the AdeS protein of the reference strain.

**Figure 3 ijms-24-09525-f003:**
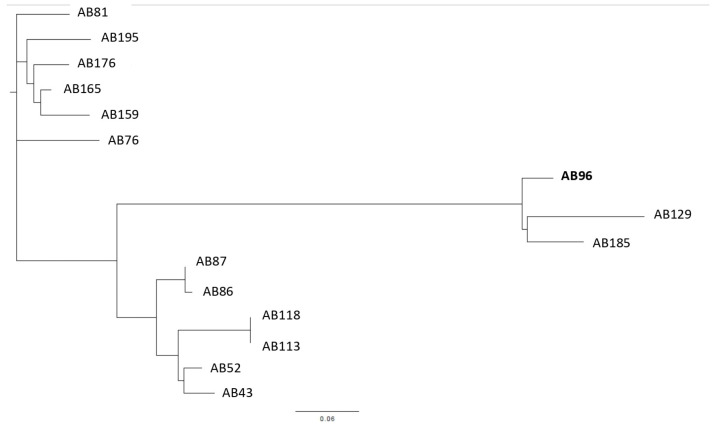
Core genome SNP-based phylogenetic tree analysis of *A. baumannii* clinical isolates (*n* = 15), where isolate *A. baumannii* 96 (AB96) was used as a reference.

**Figure 4 ijms-24-09525-f004:**
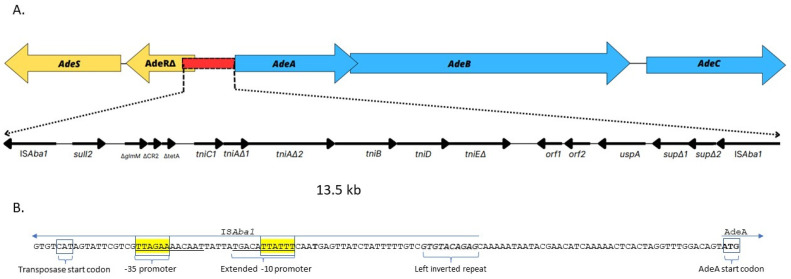
Genetic structure of *adeRS* and *adeABC* region in isolate *A. baumannii* 96. (**A**) Scheme of structure of *adeABC-adeRS* disrupted by Δ1AbaR25-fragment, i.e., 13.5 kb-long fragment of the AbaR25 resistance island, inserted between *adeR*Δ and *adeA.* (**B**) Nucleotide sequence of region upstream of *adeA* containing the promotor of ISAba1 which flanks Δ1AbaR25-fragment. The genes and open reading frames are shown by labelled arrows, and the arrowheads indicate the direction of transcription.

**Table 1 ijms-24-09525-t001:** Effect of CCCP and PAβN on the carbapenem MIC values among the *bla*_CHDL_-carrying *A. baumannii* isolates (*n* = 14).

Groups of Isolates Carrying the Following Genes (*n* = 14) [9]		Isolate	MIC (mg/L)					
			IMP ^a^	IMP + CCCP	IMP + PAβN	MEM	MEM + CCCP	MEM + PAβN
*bla* _OXA-51-like_	IS*Aba3*-*bla*_OXA-58-like_	AB43	16	**1** ^b^	8	64	32	**16**
	IS*Aba1*-*bla*_OXA-23-like_	AB87	16	8	**4**	32	16	**8**
		AB92	16	8	8	32	16	**8**
		AB96	32	16	16	32	16	**8**
		AB111	32	16	16	64	32	**16**
		AB113	32	16	**8**	64	32	**8**
		AB118	32	16	16	64	32	**16**
		AB119	16	8	16	64	32	**16**
		AB177	16	8	16	64	32	**16**
	*bla* _OXA-24-like_	AB37	32	32	32	128	64	**32**
		AB81	64	32	**16**	128	128	**32**
		AB153	32	**8**	16	64	32	32
		AB165	32	16	16	128	64	**32**
		AB195	32	16	16	128	128	**16**

^a^ IMP, imipenem; MEM, meropenem; CCCP, an efflux pump inhibitor Carbonyl cyanide 3-chlorophenylhydrazone; PAβN, an efflux pump inhibitor Ph-Arg-β-naphthylamide; AB, *Acinetobacter baumannii*. ^b^ At least a 4-fold MIC carbapenem reduction in the presence of EPI is marked in bold face.

**Table 2 ijms-24-09525-t002:** Characteristics of *A. baumannii* isolates (*n* = 15) selected for further analysis focusing on expression levels of *adeB*, *adeR and adeS* genes, as well as on changes in efflux pump proteins and their regulators in relation to the effect of PAβN on carbapenem MIC values.

Isolate Groups Producing Acquired CHDL [9]	Isolate	MIC MEM ^a^/x-Fold Reduction with PAβN	MIC IMP/x-Fold Reduction with PAβN	CA Result (Time in Minutes) [9]	Fold Upregulation ^b^			Mutation ^c^	
					*adeB*	*adeR*	*adeS*	AdeR	AdeS
OXA-58-like (*n* = 2)	AB43	64/4	16/none	Uninterpretable (120)	11	2	1	V120I, A136V	L172P, G186V, N268H, Y303F, V348I, G356N, H357N
	AB52	8/none	16/none	Uninterpretable (120)	21	1	1	V120I, A136V	L172P, G186V, N268H, Y303F, V348I, G356N, H357N
OXA-23-like (*n* = 7)	AB86	16/none	16/none	Positive (45)	11	2	1	V120I, A136V	L172P, G186V, N268H, Y303F, V348I, G356N, H357N
	AB87	32/4	16/4	Positive (45)	11	3	1	V120I, A136V	L172P, G186V, N268H, Y303F, V348I, G356N, H357N
	AB96	32/4	32/none	Positive (25)	67	1	1	V120I, A136V, **IS*Aba1* insertion**	L172P, G186V, N268H, Y303F, V348I, G356N, H357N **deletion: H102, G103**
	AB113	64/8	32/4	Positive (45)	21	1	1	V120I, A136V	L172P, G186V, N268H, Y303F, V348I, G356N, H357N
	AB118	64/4	32/none	Positive (60)	10	2	1	V120I, A136V	L172P, G186V, N268H, Y303F, V348I, G356N, H357N
	AB129	8/none	8/none	Uninterpretable (120)	49	1	1	V120I, A136V, **IS*Aba1* insertion**	L172P, G186V, N268H, Y303F, V348I, G356N, H357N **deletion: H102, G103**
	AB185	8/none	16/none	Uninterpretable (120)	58	1	1	V120I, A136V, **IS*Aba1* insertion**	L172P, G186V, N268H, Y303F, V348I, G356N, H357N **deletion: H102, G103**
OXA-24-like (*n* = 6)	AB76	128/none	64/none	Positive (45)	5	2	1	V120I, A136V	L172P, G186V, N268H, Y303F, V348I, G356N, H357N
	AB81	128/4	64/4	Positive (35)	16	2	1	V120I, A136V	L172P, G186V, N268H, Y303F, V348I, G356N, H357N
	AB159	32/none	32/none	Positive (55)	15	2	1	V120I, A136V	L172P, G186V, N268H, Y303F, V348I, G356N, H357N
	AB165	128/4	32/none	Positive (25)	14	4	3	**N115T**, V120I, A136V	L172P, G186V, N268H, Y303F, V348I, G356N, H357N
	AB176	128/none	64/none	Positive (5)	13	2	1	V120I, A136V	L172P, G186V, N268H, Y303F, V348I, G356N, H357N
	AB195	128/8	32/none	Positive (100)	14	1	1	V120I, A136V	L172P, G186V, N268H, Y303F, V348I, G356N, H357N

^a^ MEM, meropenem; IMP, imipenem; PAβN, an efflux pump inhibitor Ph-Arg-β-naphthylamide; CA, CarbAcineto NP test; AB, *Acinetobacter baumannii*. ^b^ Expression level of efflux pump gene which was at least a 3-fold higher in comparison to the control strain *A. baumannii* ATCC 17978 (expression = 1) was considered as overexpression. ^c^ Unique AdeS and AdeR amino acid sequence changes obtained in only some of the 15 isolates are marked in bold face.

**Table 3 ijms-24-09525-t003:** Sequences of primers used for the efflux pump genes’ amplification by classic PCR and for the analysis of efflux pump gene and its regulatory genes’ expression by qPCR.

Target Gene	PCR Type		Sequence (5′→3′)	Reference
*adeB*	qPCR	F ^a^	GGATTATGGCGACTGAAGGA	[57]
		R	AATACTGCCGCCAATACCAG	
*adeS*		F	GAATTCACTCCGCCGAAATGT	This study
		R	AACTCATGTGCGATAGCTGC	
*adeR*		F	TGCACTAGAGCGAACCGTAG	This study
		R	CTATATCCCACGCCACGCAC	
*fabD*		F	CCAGTATTGCTTTATGGCG	This study
		R	TGCAACTAAAGCGCTGTATTC	
*proC*		F	CTGTCGAACAAATTCGTCAA	This study
		R	CGTAGAACATTTGCCAGAACTT	
*adeB*	Classic PCR	F	TTAACGATAGCGTTGTAACC	This study
		R	TGAGCAGACAATGGAATAGT	
*abeM*		F	GCAACATCCATTTTACAGTG	This study
		R	TTGTTCACGGCCTAAAAGA	

^a^ F, forward; R, reversed.

## Data Availability

The complete whole genome sequence of clinical isolate *A. baumannii* 96 obtained from additional long-read WGS analysis is available at the NCBI BioProject repository (Submission ID: SUB12923155, BioProject ID: PRJNA939738). The whole genome datasets of the 15 strains analysed during the current study were previously deposited in the NCBI GeneBank, and are available at the NCBI BioProject repository (Submission ID: SUB9082120, BioProject ID: PRJNA701882).

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
