# Peer review of "Efflux-Related Carbapenem Resistance in Acinetobacter baumannii Is Associated with Two-Component Regulatory Efflux Systems’ Alteration and Insertion of ΔAbaR25-Type Island Fragment"

_ijms, 2023, doi:10.3390/ijms24119525_

Round 1

Reviewer 1 Report

The title clearly describes the study design.

The abstract sums up the main contents of the work with coherence and effectiveness.

In the text, there is a short summary of the past and current literature relevant to the aims of the study.

Methods

The methods section reports most of the information about the study. However, some information needs to be added.

1. The study design is missing. Please add in the Methods the type of study along with inclusion, and exclusion criteria.

2. Lines 411-415: “All studied 411 strains were characterized (clinical material type, antimicrobial susceptibility determination, phenotypic carbepenemase detection, presence of genes encoding carbapenemases, genetic relatedness between strains by PFGE method, MLST sequence type) in our previous paper.” Could the Authors add also in this article, the clinical material types of the isolates? It would be interesting to know whether there was a difference between different types of isolates and the MIC reduction.

3. In the methods, the Authors should add the specific names of the hospitals which were included in the study. Also please add how many isolates from each hospital were included in the study.

Results

4. Did the Authors test resistance to colistin of A. baumannii isolates? If yes, please add.

5. Did the Authors study also the MIC reduction of other resistant antibiotics not only carbapenems? It would be interesting to see for isolates who had a MIC reduction, if they had a reduction of the MIC also for other resistant antibiotics.

Discussion

6. Limitation of this study is missing. Please add limits of the study in the Discussion part. For example, low number of isolates. Although is a multicenter study, this is not a multinational one, therefore resistance patterns and the response of MIC reduction patterns may be influenced by local epidemiology.

Author Response

The title clearly describes the study design.

The abstract sums up the main contents of the work with coherence and effectiveness.

In the text, there is a short summary of the past and current literature relevant to the aims of the study.

Methods

The methods section reports most of the information about the study. However, some information needs to be added.

  1. The study design is missing. Please add in the Methods the type of study along with inclusion, and exclusion criteria.

Our answer: Corrected.

According lines 115-122 at the end of the Introduction section, the study design was ”to determine the impact of the most important efflux system - AdeABC and additionally AbeM pump on carbapenem resistance in clinical isolates of acquired blaCHDL genes-carrying A. baumannii. The goal was to be achieved by basing on phenotypic (susceptibility testing to carbapenems with efflux pump inhibitors (EPIs)) and molecular methods (determining expression levels of adeB pump gene with adeSR regulatory genes, whole genome sequencing and analyzing changes of the following genes: abeM, baeSR and adeSRABC, with special attention to genetic environment of the latter).”

The research conducted in this manuscript are not epidemiological studies, nor are A. baumannii isolated from specific patients or from selected clinical materials. The clinical samples were part of the routine diagnostic procedure in microbiology laboratories.

The only criterion for the inclusion of the strain in this study was resistance to imipenem and carriage the blaCHDL genes relevant for carbapenem resistance. Information about these strains is provided in section 4.1.

Moreover, as recommended by the Reviewer, we added the sentence:

All strains carried blaCHDL genes relevant for carbapenem resistance, including 59 isolates with acquired blaCHDL genes and the other 2 isolates with ISAba1-blaOXA-51-like genes [9]. Epidemiological data of isolates and the blaCHDL gene groups they possess are presented in Table S1 in the Supplementary Materials.” – line 419-422 at the Materials and Methods, in section 4.1.

  1. Lines 411-415: “All studied 411 strains were characterized (clinical material type, antimicrobial susceptibility determination, phenotypic carbepenemase detection, presence of genes encoding carbapenemases, genetic relatedness between strains by PFGE method, MLST sequence type) in our previous paper.” Could the Authors add also in this article, the clinical material types of the isolates? It would be interesting to know whether there was a difference between different types of isolates and the MIC reduction.

Our answer: Corrected.

According to section 4.1., in this manuscript we studied only a collection of 61 non-repetitive imipenem-insensitive A. baumannii isolates. Thus, 411 strains were not tested.

On line 411 we added the information “from patients hospitalized in one tertiary hospital in Warsaw, Poland. The isolates were recovered from following clinical specimens: respiratory tract samples (n=15), wound swabs (n=15), urine samples (n=15), rectal swab (n=4), blood samples (n=3), fistula swabs (n=3), stoma swab (n=3), and one isolate from each, peritoneal fluid, surgical drain swab and catheter tip.On line 421 we added the sentence "Epidemiological data of isolates and the blaCHDL gene groups they possess are presented in Table S1 in the Supplementary Materials". Thus, we added to Table S1 the following data: the year of isolation and the clinical material from which A. baumannii was isolated.

  1. In the methods, the Authors should add the specific names of the hospitals which were included in the study. Also please add how many isolates from each hospital were included in the study.

Our answer: Corrected.

On line 411 we added the information “from patients hospitalized in one tertiary hospital in Warsaw, Poland.

Results

  1. Did the Authors test resistance to colistin of A. baumannii isolates? If yes, please add.

Our answer: The characterization of a collection of the 61 A. baumannii isolates, including colistin susceptibility data, was presented in a previous publication. All isolates were susceptible to colistin. Information on colistin susceptibility is not included in this manuscript because (I) as the title states, the manuscript deals with efflux-related carbapenem resistance and (II) colistin is not a substrate for efflux pumps.

  1. Did the Authors study also the MIC reduction of other resistant antibiotics not only carbapenems? It would be interesting to see for isolates who had a MIC reduction, if they had a reduction of the MIC also for other resistant antibiotics.

Our answer: NO. We focused only on the contribution of efflux pumps to carbapenem resistance.

Discussion

  1. Limitation of this study is missing. Please add limits of the study in the Discussion part. For example, low number of isolates. Although is a multicenter study, this is not a multinational one, therefore resistance patterns and the response of MIC reduction patterns may be influenced by local epidemiology.

Our answer: According to section 4.1., in this manuscript we studied only a collection of 61 non-repetitive imipenem-insensitive A. baumannii strains isolated in the period between 2009-2014 from patients hospitalised in one tertiary hospital in Warsaw, Poland. The only criterion for the inclusion of the strain in this study was resistance to imipenem and carriage the blaCHDL genes relevant for carbapenem resistance. The research conducted in this manuscript are not epidemiological studies, nor are A. baumannii isolated from specific patients or from selected clinical materials. The clinical samples were part of the routine diagnostic procedure in microbiology laboratories.

The objective of this study was to determine the impact of the most important efflux system - AdeABC and additionally AbeM pump on carbapenem resistance in clinical isolates of acquired blaCHDL genes-carrying A. baumannii.

In the Discussion the limits of this study were written: “61 isolates used in this study” (line 286) and “The contribution of efflux to carbapenem resistance was investigated in a subset of these strains in this study” (lines 288-289).

Submission Date

12 May 2023

Date of this review

21 May 2023 17:44:2

Reviewer 2 Report

ijms-2421728

In this study, the contribution of efflux mechanism to carbopenem resistance in 61 acquired blaCHDL genes-carrying Acinetobacter baumannii clinical strains isolated in Warsaw, Poland, was investigated.

For the first time, the involvement of the insertion of ΔAbaR25-type resistance island fragment with ISAba1 element upstream the efflux operon in the carbapenem resistance of A. baumannii clinical isolates was reported in this study.

This study also revealed that AbaR25 resistance island are the reservoir of resistance genes and they may be transferred to another part of the bacterial genome.

The manuscript contains novelty, it is well written and may provide insights for future research.

Author Response

ijms-2421728

In this study, the contribution of efflux mechanism to carbopenem resistance in 61 acquired blaCHDL genes-carrying Acinetobacter baumannii clinical strains isolated in Warsaw, Poland, was investigated.

For the first time, the involvement of the insertion of ΔAbaR25-type resistance island fragment with ISAba1 element upstream the efflux operon in the carbapenem resistance of A. baumannii clinical isolates was reported in this study.

This study also revealed that AbaR25 resistance island are the reservoir of resistance genes and they may be transferred to another part of the bacterial genome.

The manuscript contains novelty, it is well written and may provide insights for future research.

Our answer: We are very grateful to Reviewer No. 2 for her/his thorough evaluation of our manuscript.

Submission Date

12 May 2023

Date of this review

20 May 2023 21:46:11